# Effects of Aluminum Addition on Microstructure and Properties of TiC-TiB_2_/Fe Coatings In Situ Synthesized by TIG Cladding

**DOI:** 10.3390/ma16144935

**Published:** 2023-07-11

**Authors:** Ning Ma, Xiao Xiao, Di Zhao, Danqing Yin, Keke Zhang

**Affiliations:** 1School of Materials Science and Engineering, Henan University of Science and Technology, Luoyang 471023, China; xiaoxiaonov@163.com (X.X.); yindanqing@hotmail.com (D.Y.);; 2Provincial and Ministerial Co-Construction of Collaborative Innovation Center for Non-Ferrous Metal New Materials and Advanced Processing Technology, Luoyang 471023, China; 3School of Materials Science and Engineering, Harbin Institute of Technology, Harbin 150006, China

**Keywords:** metal matrix composites (MMCs), TiC-TiB_2_/Fe coatings, Al addition, microstructure, wear resistance

## Abstract

This study focuses on the synthesis of TiC-TiB_2_/Fe coatings with varying amounts of aluminum (Al) using tungsten inert gas (TIG) cladding and investigates the impact of Al addition on microstructure refinement and performance enhancement of the coatings. The coatings were prepared on a mild steel substrate using TIG cladding. X-ray diffraction (XRD) analysis revealed the presence of TiC, TiB_2_, Al_x_Ti, and Al_x_Fe phases in the coatings. Scanning electron microscopy (SEM) images showed that the addition of Al improved the microstructure, reducing defects and enhancing the distribution of reinforcing phases within the coatings. The particle size of the reinforcing phases was significantly refined by the addition of Al. The micro-hardness of the coatings was significantly higher than that of the substrate, with the maximum micro-hardness of the coating reaching 955.5 ± 50.7 HV0.1, approximately six times that of the substrates. However, excessive Al addition led to a reduction in hardness due to a decrease in the quantity of hard phases. The wear tests showed that all the coatings had lower wear loss compared to the substrate material, with the wear loss initially decreasing and then increasing with the increasing Al content. Samples with a 28.57 wt.% Al addition exhibited the best wear resistance, with approximately 16.8% of the wear volume loss compared to mild steel under the same testing conditions, attributed to the optimal combination of reinforcement phase quantity and matrix properties.

## 1. Introduction

By combining metal matrices with ceramic reinforcements, metal matrix composites (MMCs) provide a promising approach for developing advanced components with improved mechanical strength, hardness, and wear resistance [1], making them suitable for wear-resistant and anti-friction applications [2]. In the form of coatings, MMCs can effectively safeguard components from wear [3], erosion [4], and other forms of degradation [5], thereby significantly improving the lifespan and performance of the underlying material [6].

The volume fraction and size of the reinforcing phase are key factors influencing the performance of MMCs. Rahmani et al. [7] found that the volume fractions of SiC reinforcement significantly affect the relative density, micro-hardness, and strength of Mg-SiC MMCs. Increasing the SiC content improves the hardness and strength of the material but decreases the relative density [7]. Additionally, grain refinement plays a beneficial role in further enhancing the performance of MMCs. Rahmani et al. [8] also reported that magnesium composites reinforced with five volume percent of ZrO_2_ and TiO_2_ nanoparticles exhibited tensile strengths 2.5 and 2.1 times higher than that of unreinforced magnesium. Majzoobi et al. [9] investigated the effect of pre-compaction on the mechanical properties of Mg/SiC nanocomposites prepared through dynamic compaction, and the results indicated that there existed an optimum pre-compaction pressure, which varies depending on the matrix type, reinforcing particles, and compaction loading rate. Zhang et al. [10] developed a novel method by adding trace in situ synthesized nano-(TiC + TiB_2_) particles to forged H13 steel, resulting in refined grains and laths martensite, increased retained austenite and carbides, and controlled carbide segregation. This simultaneous improvement in strength, plasticity, and toughness offers a new approach for bonding ceramic nanoparticles to steel and suggests a novel idea for the development of high-performance and cost-effective steel.

Furthermore, in the pursuit of fabricating composite coatings with exceptional properties, the development of hybrid composite (HC) coatings incorporating two or more different types of reinforcing phases has emerged as an active area of research and innovation in material science and engineering [11]. Researchers have explored the use of multiple reinforcing phases in MMCs, such as TiC-TiB_2_, Al_2_O_3_-TiB_2_, and Al_2_O_3_-TiB_2_-TiC. These combinations offer the potential to synergistically enhance the properties of the composite coatings. Anbarasan et al. [12] developed a TiB_2_ and TiC hybrid-reinforced titanium matrix composite produced through powder metallurgy. Tijo et al. in situ synthesized TiC-TiB_2_ coating on Ti-6Al-4V alloy by tungsten inert gas (TIG) cladding, and investigated the microstructure evolution [13] and mechanical performances [14] of the coatings. Philip et al. [15] successfully fabricated Al_2_O_3_-TiB_2_ MMCs on AA6061 aluminum alloys fabricated by in situ reaction of boric oxide and Ti metallic powder. Alvar et al. [16] developed Al_2_O_3_-TiB_2_ nanocomposite coatings on pure titanium and found that the coatings exhibited good corrosion resistance and biocompatibility. Li et al. [17] developed Al_2_O_3_-TiB_2_-TiC ceramic coatings on the surfaces of carbon steel substrates by laser cladding, demonstrating excellent wear resistance and oxidation resistance. Xu et al. [18] fabricated Al_2_O_3_-TiB_2_-TiC/Al metal matrix composite coatings by atmospheric plasma spraying of SHS powders. The resulting coatings exhibited a significant increase in microhardness and wear resistance compared to the MB26 substrate [18]. Among the various MMCs, the Fe-based MMC coatings are widely utilized in the wear resistance applications due to their lower manufacturing costs, high stiffness and elastic modulus, and good compatibility with various substrate materials. According to Chen et al. [19], who focused on metal parts remanufacturing, the development of Fe-based coatings reinforced with TiC and TiB_2_ has shown promising results. These in situ formed ceramic-reinforced composite coatings have demonstrated their potential as an ideal choice for remanufacturing damaged parts, effectively improving their wear resistance [19]. Yu et al. conducted a study on the fabrication of in situ synthesized TiB_2_ and TiC particle-reinforced composite coatings on shaft parts via laser cladding, which demonstrated a 50% reduction in wear resistance while meeting the machining quality requirements [20].

Notably, TiC and TiB_2_ are of particular significance in the preparation of Fe-based composite coatings due to their excellent properties, including high hardness, high melting point, low density, excellent corrosion resistance, and good thermal stability [21]. By using B_4_C and Ti as raw materials, in situ fabrication of TiC- and TiB_2_-reinforced MMCs can be achieved. This in situ fabrication approach allows for the formation of TiC and TiB_2_ particles within the MMCs, leading to improved interfacial bonding, enhanced dispersion, and reduced porosity [22]. Additionally, some researchers have found that the addition of Al elements can have beneficial effects in promoting the reaction between B_4_C and Ti and refining TiC and TiB_2_ grains. Shen et al. [23] developed a TiC_x_-TiB_2_/Al composite coating and found that the addition of Al to the Ti-B_4_C system significantly promoted the reaction between B_4_C and Ti. Wang et al. [24] observed that increasing the aluminum content from 20% to 40% in the self-propagating high-temperature synthesis reaction of the Al-Ti-C system in molten magnesium resulted in reduced TiC particle sizes and a uniform distribution of TiC particles in the matrix. Cui et al. [25] fabricated TiC-TiB_2_-NiAl composites by the SHS method using Ti, B_4_C, Ni, and Al powders. They observed that the size of both TiC and TiB_2_ particles in the TiC-TiB_2_-NiAl composites was refined and that the density and compressive strength of these composites increased [25].

However, to the best of our knowledge, there is a lack of relevant reports on the effects of Al addition on the microstructure and properties of TiC-TiB_2_/Fe coatings prepared by tungsten inert gas (TIG) cladding. Therefore, the aim of this study was to in situ synthesize a series of TiC-TiB_2_/Fe coatings with varying amounts of Al addition using TIG cladding from a FeTi-B_4_C system and investigate how the addition of Al affects the microstructure refinement and performance enhancement of the coatings.

## 2. Materials and Methods

The TiC-TiB_2_/Fe coatings were prepared on the surface of Q235 steel (a commonly used mild steel in China, equivalent to ASTM A36 steel in the United States) by TIG cladding. TIG cladding was performed using a tungsten electrode and argon gas as the heat source to melt the pre-placed powder and form the coating. The cladding was carried out using a YC-300WX4 N-type TIG welding machine with direct current electrode negative (DCEN) polarity. A tungsten electrode with a diameter of 2.0 mm was used, and the welding current was set to 60 A. Air cooling was employed for the cooling process. Argon gas with a purity of 99.9% and a flow rate of 10 L/min was used to shield and stabilize the arc. The schematic diagram of TIG cladding is shown in Figure 1 [13]. The scan speeds were set to 2.0 mm/s. Before cladding, the surface of the substrates was polished with 800-mesh emery paper and cleaned with acetone. The surface roughness (Ra) was approximately 0.6 μm. Mixtures of Al, FeTi70, and B_4_C powders were used as precursors for coating preparation. The main chemical components of the powders are shown in Table 1, and the particle sizes of all powders were −300 mesh. The component ratios of the precursor powders and the corresponding sample numbers are listed in Table 2. The mass ratio of FeTi70 and B_4_C powders was fixed at 4:1, which is approximately equal to the molar ratio of complete reaction of 3Ti + B_4_C = 2TiB_2_ + TiC. Based on this ratio, the ratio of Al powder was increased. Firstly, the powders were mixed using a ball milling technique for three hours. Subsequently, the blended powders were pre-placed on the surface of Q235 steel with organic binder to give a thickness of approximately 1.0–1.5 mm. The samples were then set in a ventilated place at room temperature for 24 h and subsequently placed into a vacuum drying oven at 80 ℃ for 1 h.

The microstructure and chemical composition of the samples were analyzed by a JSM-5610 LV scanning electron microscope (SEM), which was attached with an energy dispersive spectrometer (EDS). Phase constituents of the coatings were identified by a D8-ADVANCE X-ray diffractometer (XRD) with an accelerating voltage of 40 KV and a current of 40 mA. The XRD scanning speed was set at 2°/min in the range of 20–90°. The Vickers microhardness of the coatings along the depth of the cross-section was measured using an MHV-2000 microhardness tester (Beijing TIME Shuncheng Technology Co., Ltd., Beijing, China) with a load of 100 g and a loading time of 10 s [26]. The microhardness value was from the average of 5 measurements. According to the national standard of the People’s Republic of China, GB 12444.2-90 [27], “Metallic materials—Wear tests Block-on-ring wear test,” dry sliding wear tests were carried out at room temperature. The tests were conducted using a block-on-ring wear tester (MM-200, Zhangjiakou, China) without the use of any lubrication [26]. A quenched bearing steel ring with a hardness of 65 HRC was used as the wear couple. The ring had an outer diameter of 40 mm and a width of 10 mm. Prior to the tests, the samples were machined to a size of 15 mm × 7 mm × 7 mm and were ground using silicon carbide sandpapers. The wear tests were conducted using a normal load of 98 N, a sliding speed 0.31 m/s, and a sliding distance of 2260 m. After the tests, the width of the wear track *b* was measured using a vernier caliper with a precision of 0.01 mm. The volume of wear volume loss *V* can be obtained using the following formula:(1)$$V={\frac{Dt}{8}}^{2}\left[2\sin^{-1}\frac{b}{D}-\sin(2\sin^{-1}\frac{b}{D})\right]$$
where, *V* is the volume of wear loss, mm^3^;*D* is the outer diameter of the ring, mm;*t* is the width of the specimen, mm;*b is* the width of the wear track, mm.

The wear resistance of the coating specimens is measured by the wear rate, denoted as *W_r_*. The calculation formula is as follows:(2)$$W_{r}=\frac{V}{F\times S}$$
where, *W_r_* is the wear rate of the specimen, mm^3^/Nm;*V* is the wear volume loss, mm^3^;*F* is the applied load, N;*S* is the sliding distance, m.

During the wear test, the friction torque is recorded, and the coefficient of friction (COF) of the coating can be calculated based on the friction torque. To minimize random errors, the final values of wear volume loss and COF were obtained by conducting three tests under the same conditions. Subsequently, the worn surface of the samples was examined using scanning electron microscopy.

## 3. Results and Discussion

### 3.1. X-ray Results of Coatings

Figure 2 shows the XRD patterns of the coatings with varying Al contents. The patterns revealed that TiC and TiB_2_ peaks were detected in all samples, indicating that the TiC and TiB_2_ particles were in situ synthesized from the FeTi70 and B_4_C powders during the cladding process. When Al powders were added, Al_x_Ti and Al_x_Fe phases were detected in the coatings. As the Al content increased from 16.67% to 44.44%, the Al_x_Fe phases changed from Al_2_Fe_3_ (PDF#45-0982) to Al_13_Fe_4_ (PDF#47-1420) phases. Sample S3 had a higher relative diffraction peak intensity of TiB_2_ than samples S2 and S4, indicating that the content of TiB_2_ in sample S3 was relatively higher than that in samples S2 and S4. Additionally, intermediate phase TiB appeared in the coatings, likely due to the molten pool undergoing an incompletely non-equilibrium reaction under the extremely rapid solidification process. Although the diffraction peaks were not salient, small amounts of Fe_x_(C,B) phases were also detected in all samples as reaction products. Additionally, the peaks of TiB_2_ and the matrix (α-Fe, AlxFe) overlap significantly, which is also observed in the references [21,28,29]. This situation prevents us from performing certain XRD quantitative analyses, such as estimating the content ratio of different phases based on peak intensities or using Scherrer’s equation to calculate the grain size by analyzing the peak broadening of XRD peaks.

### 3.2. Microstructure of Coatings

Figure 3 shows the SEM micrographs near the interface between the substrate and the coatings in sample S1 (a) and sample S3 (b) at a lower magnification (500×). The microstructures at a lower magnification in samples S2, S3, and S4 were similar. Therefore, sample S3 was chosen as a representative sample with the addition of Al element to compare with sample S1, which did not have Al element addition.

Several pores and slag inclusion defects can be observed in sample S1. However, in sample S3, the base metal and coating exhibited a good metallurgical bond, with no apparent cracks, pores, or other defects. This indicates that the addition of Al element improved the microstructure and reduced the coating defects. This improvement can be attributed to the low melting point of Al (660 °C). During the cladding process, Al melted first, forming a molten pool. The longer holding time of the molten pool, resulting from the addition of Al, enhanced the metallurgical reaction [30]. As a result, gases and slag inclusions had sufficient time to rise to the top of the molten pool.

Figure 4 presents a higher magnification cross-section micrograph of the coatings with varying Al contents. In Figure 4a, it is evident that the reinforcing phases of sample S1 were not uniformly distributed within the matrix. However, the addition of Al element improved the distribution of reinforcing phases in the coatings, as illustrated in Figure 4b–d. The coatings exhibited three main types of reinforcing phases. One type was rectangular in shape (point “1” in Figure 4a), another was granular (point “2” in Figure 4b), and the third type was network-shaped (point “3” in Figure 4). Similar typical microstructural features have also been observed in the literature [21].

Figure 5 presents the EDS analysis results of these three reinforcing phases. In Figure 5a, the main elements at point “1” were Ti and B, suggesting that the rectangular-shaped particles were TiB_2_ or TiB. Figure 5b indicates that the granular particles were composed of Ti and C, implying that they were likely TiC. Additionally, Figure 5b reveals that the network-shaped particles were rich in Fe, C, and B elements, indicating the formation of Fe_x_ (C,B) compound in the coating. The elements of the matrix (point “4” in Figure 4) in the coatings were also examined, as shown in Figure 5d, and mainly comprised Fe, Al, and Ti. Combined with the XRD analysis results, it can be inferred that the matrix of the coatings consists of intermetallic compounds such as Al_x_Ti and Al_x_Fe.

The results depicted in Figure 4 demonstrate that the particle size of the reinforcing phase was significantly refined by the addition of Al. In particular, sample S3 exhibited a uniformly distributed reinforcing phase with smaller particle size compared to the other samples. This can be primarily attributed to the significant effects of Al on the reaction behaviors and phase compositions [23].

Upon adding a small amount of Al to the precursor, the formation of Al_x_Ti and Al_x_Fe intermetallic compounds occurred initially. These intermetallic compound layers surrounded the B_4_C particles, facilitating their contact with Al_x_Ti and reducing the atomic diffusion distance for subsequent reactions. According to references [23,31], the diffusion of Al into the B_4_C crystal structure may have played a role in breaking the bonds between boron and carbon, thereby promoting the dissociation of B_4_C. Furthermore, some researchers have reported that the Al_x_Ti phase acts as a master alloy in the Al-Ti-C system, leading to the refinement of TiC and TiB_2_ particle sizes [32,33]. However, when the addition of Al powder increased to 44.4%, the amount of strengthening phases in the coatings was noticeably reduced, as observed in Figure 4d.

### 3.3. Micro-Hardness Analysis of the Coatings

The micro-hardness measurements along the cross-section of the specimens are presented in Figure 6. The micro-hardness of the coatings was significantly higher than that of the substrate, with the maximum micro-hardness of the coating reaching 955.5 ± 50.7 HV0.1 (S1), approximately six times that of the substrates. This increase in hardness can be attributed to the presence of hard reinforcement phases synthesized in situ during the cladding process. From Figure 6, it is evident that the average micro-hardness of sample S1 was the highest, followed by sample S3 (826.1 ± 59.9 HV0.1) and S2 (769.5 ± 51.3 HV0.1), while the hardness of sample S4 was relatively lower (582.5 ± 39.7 HV0.1). The hardness values of these coatings were comparable to the TiC-TiB2-reinforced MMC coatings reported in the literature for TIG cladding and some laser cladding processes [14,19], significantly higher than the hardness of various common steel materials. However, they were lower than Wang’s laser-cladded TiC-TiB_2_/Fe coatings (approximately 1100–1300 HV), which can be attributed to the inclusion of a considerable amount of Al_2_O_3_ as an additional reinforcing phase in Wang’s samples [21].

This difference in hardness among the four coatings can primarily be attributed to the addition of aluminum, which influenced the relative amount of reinforcement phases. However, an appropriate aluminum content resulted in the refinement and uniform distribution of the reinforcement phase particles, as observed in Figure 4. Simultaneously, the matrix of the coatings transformed from α-Fe into Fe_x_Al intermetallic compounds, further contributing to the improved hardness of the coatings. Conversely, when too much aluminum was added, as seen in sample S4, the quantity of hard phases decreased significantly, leading to a noticeable reduction in hardness.

Furthermore, we observed a gradual increase in micro-hardness from the base metal to the coatings. This phenomenon can be attributed to the gradient distribution of TiC and TiB_2_ particles. The density of TiC (4.92 g/cm^3^) and TiB2 (4.52 g/cm^3^) is lower than that of Fe (approximately 7.86 g/cm^3^) [21]. As a result, these reinforcing phases have a natural tendency to migrate towards the upper zone during the cladding process.

However, the rapid solidification of the molten pool captures these reinforcing phases and incorporates them into the coatings. Consequently, the combined effects of floating and capturing lead to the gradient distribution of these hard particles. This gradient micro-hardness distribution helps reduce stress concentration at the interface, thereby improving the service life of the workpieces during operation.

### 3.4. Wear Resistance Analysis of the Coatings

Figure 7 illustrates the wear volume loss of the coatings and the substrate material (Q235 steel). It is evident that the wear loss of all the coatings is significantly lower than that of the Q235 steel. Among the coatings, S3 coating exhibits the best wear resistance, with a wear loss approximately 16.8% that of Q235 steel. This can be attributed to the high hardness of the in situ synthesized hard reinforcement particles within the coatings. These hard particles are capable of withstanding the alternating load during the dry wear test, thereby reducing the contact between the matrix and the grinding ring [34].

Figure 8 shows the wear rate and COF of the four coatings and the Q235 steel specimens. The wear rates of the coatings are in the range of 10^−6^ to 10^−5^ mm^3^ N^−1^ m^−1^. Among these four coated samples, the S3 coating, with a 28.57 wt.% Al addition, exhibited the lowest wear rate of 6.59 ± 0.88 × 10^−6^ mm^3^ N^−1^ m^−1^, while the S4 coating, with a 44.44 wt.% Al addition, exhibited the highest wear rate of 14.85 ± 1.45 × 10^−6^ mm^3^ N^−1^ m^−1^. The COF values of the four coatings are in the range of 0.4 ~ 0.5, which are close to the values reported in [21] and lower than that of Q235 steel.

As shown in Figure 7 and Figure 8, the wear loss of the coatings initially decreases and then increases with the increase in Al addition. Interestingly, although sample S1 has the highest micro-hardness value, its wear rate is not the lowest. This discrepancy can be explained by considering factors beyond hardness, such as the microstructure of the coatings. In Figure 3a, it can be observed that sample S1 exhibits some slag inclusions in the coating, which can contribute to increased wear loss. However, proper aluminum addition refines the particles of the reinforcement phases and ensures their uniform distribution, as mentioned earlier. Additionally, the addition of Al element alters the matrix of the coatings and increases the relative content of the matrix alloy, as depicted in Figure 4. The matrix material plays a crucial role in supporting the reinforcing phases. Specifically, the matrix of sample S1 is composed of α-Fe, whereas Al_x_Fe and Al_x_Ti intermetallic compounds appear in the matrix of samples S2–S4 due to the addition of Al element, as seen in Figure 2. The hardness of α-Fe is lower than that of Al_x_Fe and Al_x_Ti intermetallic compounds. Furthermore, the addition of Al element enhances the supporting effect of the matrix on the reinforcement particles. Consequently, the wear resistance of the coatings improves with the appropriate amount of Al addition.

Figure 9 presents the morphology of the worn surfaces of the coatings after dry wear tests. In Figure 9a, numerous irregular pits can be observed on the worn surface of coating S1. However, in Figure 9b–d, the worn surfaces of samples S2–S4 show minimal to no presence of small pits. This can be attributed to the addition of the appropriate amount of aluminum (Al) in the precursors, which increased the matrix metal content and facilitated the formation of Al_x_Fe and Al_x_Ti intermetallic compounds in the matrix. These compounds effectively hindered the spalling of hard particles. Additionally, grooves were observed on the worn surfaces of samples S2–S4. Notably, the grooves on the worn surface of sample S4 were deeper than those of samples S2 and S3, and traces of plastic deformation were also evident. The excessive addition of Al reduced the relative content of the reinforcing phases, leading to these observations.

Based on a comparison with previously published literature, it can be concluded that the main wear mechanisms of MMC coatings include spalling or fracture of reinforcing phase particles [26], as well as plastic erasing and removal of the matrix phase [34], such as micro-cutting or micro-plowing. In Figure 9a, the presence of irregular pits can be attributed to the spalling of hard particles. Comparing it with Figure 4a, it can be observed that coating S1 had a relatively high content of hard phases but a low content of matrix metal, which resulted in the spalling of hard particles without sufficient support and in the formation of these small pits. The grooves on the worn surfaces of samples S2–S4 are typical features of micro-cutting or micro-plowing [34]. Especially in sample S4, due to the decrease in hardness, the worn surface exhibited deeper grooves and traces of plastic deformation. Therefore, we can infer that the wear mechanisms of the coatings undergo certain changes with the quantity of reinforcing phases and the properties of the matrix materials. When the content of the binder phase is low, combined with the presence of some microstructural defects (see Figure 3), the hard particles in coating S1 lack sufficient peripheral bonding support, leading to a wear mechanism primarily characterized by spalling of hard particles followed by abrasive wear. However, due to the high hardness, the overall wear rate is still relatively low. With the addition of aluminum (Al), the observed phenomena in coating S1 are improved, and the surfaces of coatings S2 and S3 show almost no small pits. Some shallow grooves indicate the presence of abrasive wear mechanisms. Wang’s research has shown that when the content of hard particles is high, they carry a larger portion of the load and reduce the real contact area during the wear process, resulting in severe plastic deformation and adhesion [21]. However, when the content of hard particles is relatively low, such as in sample S4, there is a significant decrease in hardness, and the surface’s resistance to plastic deformation is reduced, leading to prominent features of plastic removal and a higher wear rate compared to the other three coating samples. Based on the above analysis, we can conclude that S3 exhibits the best wear resistance, benefiting from a combination of factors including reduced defects, matrix strengthening, appropriate content of reinforcing phases, and uniform refinement of the microstructure.

## 4. Conclusions

(1)TiC-TiB_2_ particles-reinforced Fe-based composite coatings were successfully synthesized in situ through TIG cladding using FeTi70, B_4_C, and Al powders. The addition of Al powders facilitated the synthesis of Al_x_Fe and Al_x_Ti phases in the coatings.(2)The inclusion of Al element in the precursors reduced the occurrence of defects in the coatings. Although the relative amount of reinforcement phases decreased with increasing Al powders, the particles of reinforcement phases were refined and exhibited uniform distribution.(3)The micro-hardness of the coatings was significantly higher than that of the substrate, with the maximum micro-hardness of the coating reaching 955.5 ± 50.7 HV0.1, approximately six times that of the substrates. However, excessive Al addition led to a reduction in hardness due to a decrease in the quantity of hard phases.(4)The appropriate addition of Al powders improved the wear resistance of the TiC-TiB_2_/Fe coatings. Sample S3, with a 28.57 wt.% Al addition, demonstrated the best wear resistance due to the optimal combination of reinforcement phase quantity and matrix properties. The wear rate of sample S3 was approximately 16.8% that of the mild steel under the same testing conditions.

## Figures and Tables

**Figure 1 materials-16-04935-f001:**
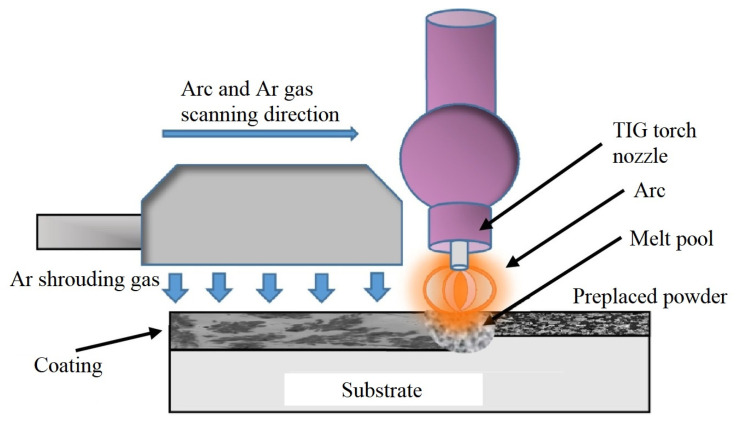
The schematic diagram of TIG cladding [13].

**Figure 2 materials-16-04935-f002:**
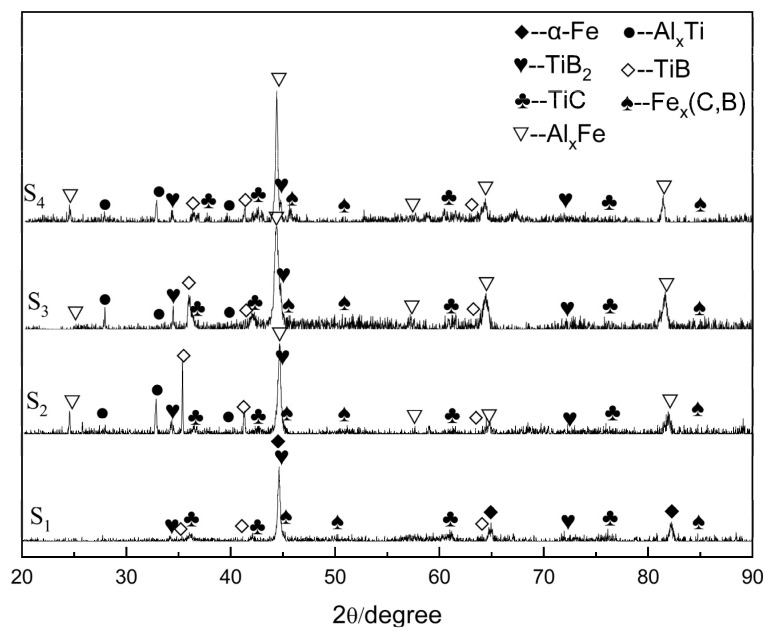
XRD diffraction patterns of coatings.

**Figure 3 materials-16-04935-f003:**
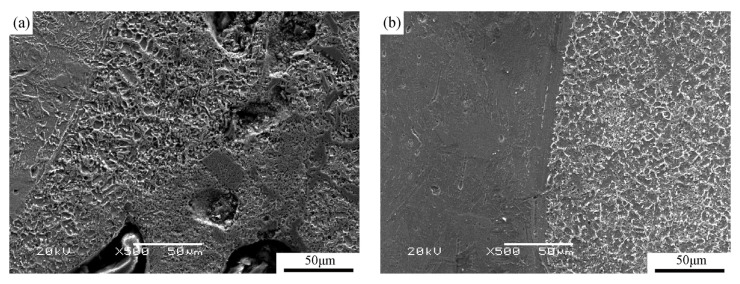
SEM micrographs of near the interface between the substrate and the coatings in the samples without Al element addition (S1) (**a**) and with Al element addition (S3) (**b**).

**Figure 4 materials-16-04935-f004:**
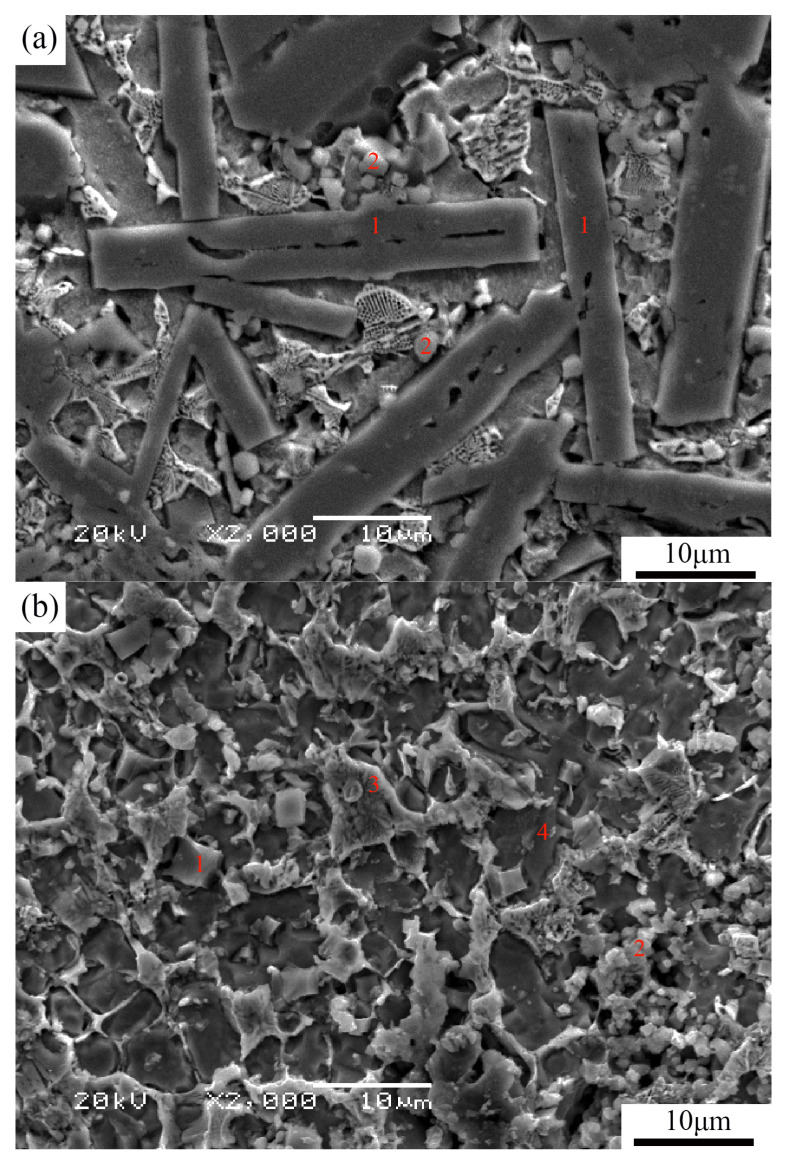
Type microstructure of the coatings with different mass fraction of Al content: sample S_1_ (**a**), sample S_2_ (**b**), sample S_3_ (**c**), and sample S_4_ (**d**). The red numbers represent four different types of microstructural features, which have been analyzed by EDS.

**Figure 5 materials-16-04935-f005:**
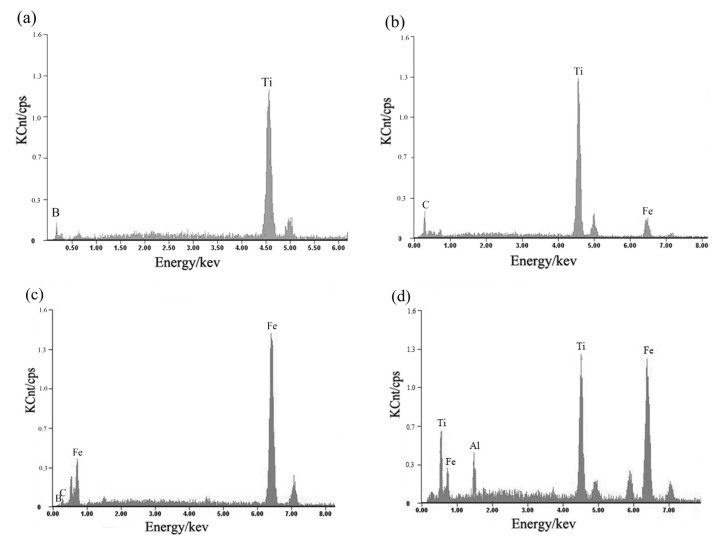
EDS element analysis results of point “1” (**a**), point “2” (**b**), point “3” (**c**), and point “4” (**d**) in Figure 4.

**Figure 6 materials-16-04935-f006:**
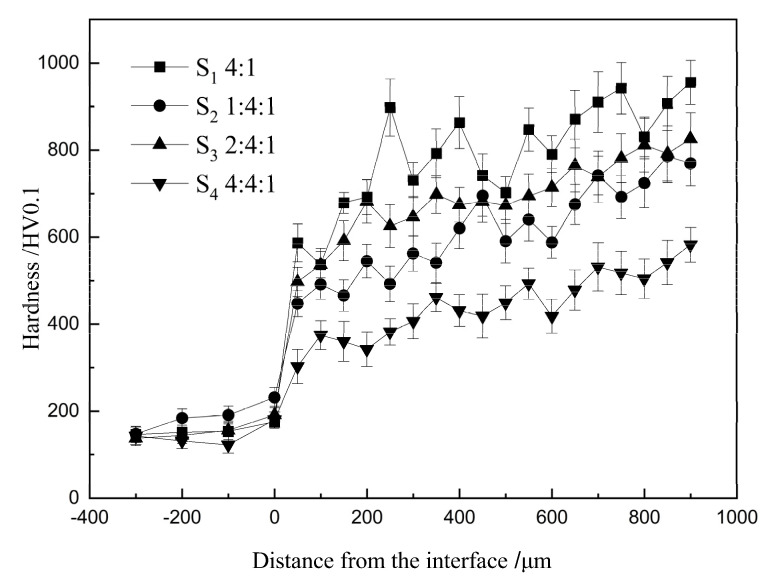
Micro-hardness profile along the cross−section of the coatings.

**Figure 7 materials-16-04935-f007:**
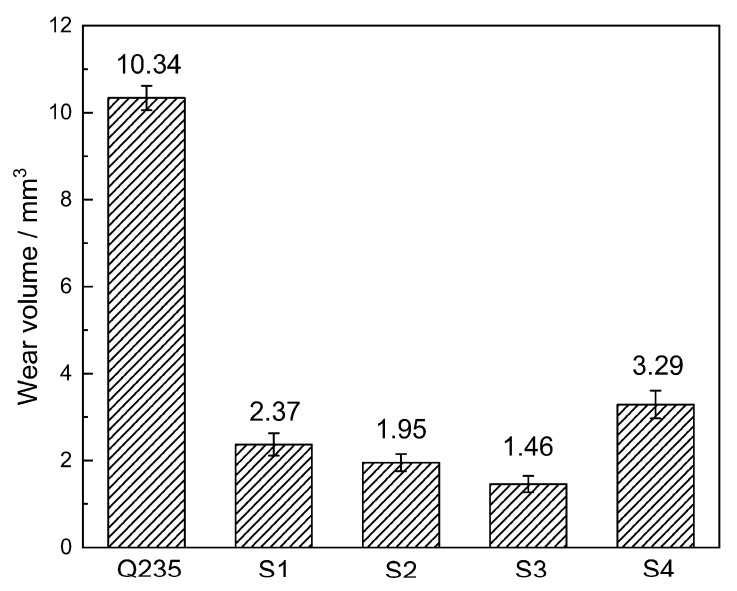
Wear volume loss of the coatings and Q235 steel.

**Figure 8 materials-16-04935-f008:**
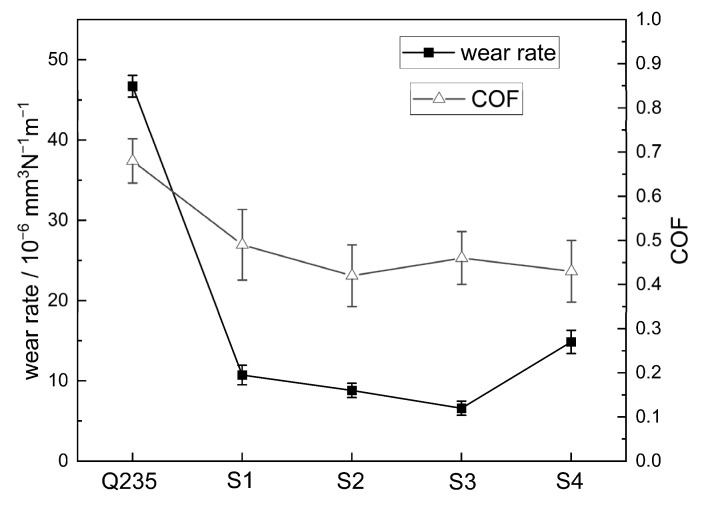
Wear rate and COF of the coatings and Q235 steel.

**Figure 9 materials-16-04935-f009:**
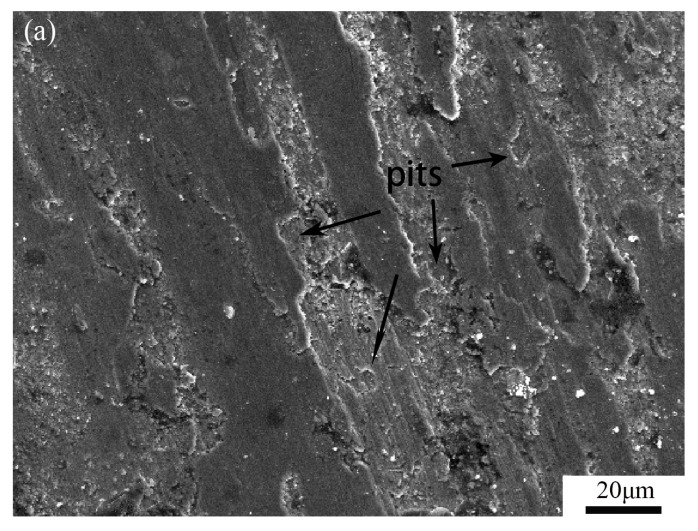
SEM image of the worn surfaces: (**a**) sample S1, (**b**) sample S2, (**c**) sample S3, and (**d**) sample S4.

**Table 1 materials-16-04935-t001:** The chemical components of precursor powders (wt.%).

Powders	Chemical Component (wt.%)					
Al	Al ≥ 99	Si ≤ 0.3	Fe ≤ 0.6	N ≤ 0.01	-	-
FeTi70	Ti 65 ~ 75	Al ≤ 0.5	Si ≤ 0.2	C ≤ 0.1	N ≤ 0.25	bal Fe
B_4_C	B_4_C ≥ 96	Mg ≤ 2.0	B_2_O_3_ ≤ 0.3	-	-	-

**Table 2 materials-16-04935-t002:** The component ratios of precursor powders (wt.%).

Sample Number	Al	FeTi70	B_4_C	Component Ratio
S1	0	80	20	0:4:1
S2	16.67	66.67	16.66	1:4:1
S3	28.57	57.14	14.29	2:4:1
S4	44.44	44.44	11.12	4:4:1

## Data Availability

Data sharing is not applicable to this article.

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
