# Peer review of "Effects of Aluminum Addition on Microstructure and Properties of TiC-TiB2/Fe Coatings In Situ Synthesized by TIG Cladding"

_materials, 2023, doi:10.3390/ma16144935_

Round 1

Reviewer 1 Report

This is a good article, with rigorous experiments. 

The bibliography must be justified by listing each reference and avoiding [9,16]. For example, Smith et al. shown .... [1].

Figure 4 EDS element analysis results of point “1”, point “2”and point “3” in Figure 3. There are 4 figures. UnclearPlease add roughness values of the substrate.

Figure 5. Discuss the size of the indentation print versus the size of the microstructure? Why desertion increases for indentation in the substrate.

Is that possible to sum up all results in a table ?

To my opinion, Sample S3, with a 28.57 wt.% Al addition shown best results with compromises of course (hardness, wear, microstructure, interface roughness). You do not discuss why this configuration is the best.

This article is therefore original. It lacks 3D surface roughness. In addition, we have no idea of the toughness at the interfaces (bending test, scratch test). Scratch tests are missing from this study, which could also have provided information on the nature of Tribological test damage.

Do you have an opinion on the influence of Al on corrosion?

Reviewer 2 Report

The presented article is well and clearly written, but there are a number of minor comments on it.

1) A short description of the TIG cladding process or a corresponding scheme should be given.

2) Designations (points) corresponding to different types of strengthening phases are not visible in fig. 3.

3) Line 168: Figure 4(c) should be written instead of figure 4(b).

4) Lines 174-175: "The results depicted in Figure 3 demonstrate that the grain size of the reinforcing phase was significantly refined by the addition of Al." It seems that here and below, instead of grains of the strengthening phase, it would be more correct to write particles of the strengthening phase.

5) The results obtained in this work should be compared with those obtained in [17] Wang, X. H., Zhang, M., Du, B. S., Li, S. Microstructure and Wear Properties of Laser Clad TiB2+TiC/Fe Composite Coating. Surf. Rev. Lett. 2012, 19, 1250052.

Reviewer 3 Report

Comments on the paper are given as follows:

1.      The abstract should be revised, it is too long, while it should contain essential conclusions.

2.      some references are missing such as the Wear test Standard, Hardness test standard, …

3.      The manuscript (Introduction and results) has a deficiency of citations to similar works published before such as Investigation Metal Matrix Composites (MMCs) physical properties in powder metallurgy methods: [1] " The effect of cold and hot pressing on mechanical properties and tribological behavior of Mg-Al2O3 nanocomposites", [2] "Determination of tensile behavior of hot-pressed Mg–TiO2 and Mg–ZrO2 nanocomposites using indentation test and a holistic inverse modeling technique", [3] "An investigation on SiC volume fraction and temperature on static and dynamic behavior of Mg-SiC nanocomposite fabricated by powder metallurgy", [4] "The effect of pre-compaction on properties of Mg/SiC nanocomposites compacted at high strain rates".

4.      The SEM and EDX for fabricated samples are needed please added to the manuscript.

5.      The XRD method is suggested for determining the grain size.

6.      Resolution of all images should be improved.

7.      Please the authors add to the manuscript the coefficient of friction and wear rate.

8.      The authors should show FESEM images for investigation of wear mechanisms.

9.      The detailed wear mechanism should be fully discussed.

10.  Please summarize the conclusion section and show the highlighted results as a list.

11.  Please discuss the result repeatability in mechanical properties characterization and explain how many tests were carried out. This is questionable for one specimen, one test. The uncertainty and errors of the experimental results must be discussed.

No comment

Reviewer 4 Report

The paper is devoted to the investigation of the effects of aluminum addition on the structure and properties of TiC-TiB2 containing coatings produced by TIG cladding

The topic is relevant and can be of interest for the wide scientific community.

The paper contains enough details, appropriate citations and coherent conclusions, although I found some points to be addressed:

The statement of the very high content of the hard phases and low content of the matrix metal in the coating S1 (Page 8, lines 260-262 and 275-276) do not fit in with the corresponding XRD pattern (Fig.  1) exhibiting the much higher intensity of Fe reflections and low-intensive reinforcing phases lines. How can you explain that?

Usage of wear mass losses as a feature to compare wear resistance of the materials with different density seems to be unconvincing.

The designations of points 1-3 in Fig. 3 can be hardly recognized. Could you improve that?

Fig. 4 consists of 4 images but there are only 3 points mentioned in the figure caption. Can you rewrite the caption?

On the whole, I recommend a minor revision.

Round 2

Reviewer 3 Report

no comment

no comment